

# Modeling the growth and sporulation dynamics of the macroalga *Ulva* in mixed-age populations in cultivation and the formation of green tides

**Uri Obolski[1,2], Thomas Wichard[3], Alvaro Israel[4], Alexander Golberg[1], and Alexander Liberzon[5]**

[1]Porter School of the Environment and Earth Sciences, Tel Aviv University, Tel Aviv, Israel
[2]School of Public Health, Tel Aviv University, Tel Aviv, Israel
[3]Institute for Inorganic and Analytical Chemistry, Friedrich Schiller University Jena, Jena, Germany
[4]Israel Oceanographic & Limnological Research Ltd. (PBC), Tel Shikmona, Haifa, Israel
[5]School of Mechanical Engineering, Tel Aviv University, Tel Aviv, Israel

**Correspondence:** Uri Obolski (uriobols@tauex.tau.ac.il) and Alexander Liberzon (alexlib@tauex.tau.ac.il)

**Abstract.** *Ulva* is a widespread green algal genus with important ecological roles and promising potential as a seagriculture crop. One of the major challenges when cultivating *Ulva* is sudden biomass disappearance, likely caused by uncontrolled and unpredicted massive sporulation. However, the dynamics of this process are still poorly understood. In this study, we propose a mathematical model describing the biomass accumulation and degradation of *Ulva,* considering the potential impact of sporulation inhibitors. We developed a differential equation model describing the time evolution of *Ulva* biomass. Our model simulates biomass in compartments of different *Ulva* "age" classes, with varying growth and sporulation rates. Coupled with these classes is a differential equation describing the presence of a sporulation inhibitor, produced and secreted by the algae. Our model mimics observed *Ulva* dynamics. We present *Ulva*'s biomass accumulation under different initial algae population, age distributions and sporulation rates. Furthermore, we simulate water replacement, effectively depleting the sporulation inhibitor, and examine its effects on *Ulva*'s biomass accumulation. The model developed in this work is the first step towards understanding the dynamics of *Ulva* growth and degradation. Future work refining and expanding our results should prove beneficial to the ecological research and industrial growth of *Ulva*.

## 1 Introduction

The genus *Ulva* (Ulvales, Chlorophyta) comprises a group of green macroalgae, which are cosmopolitan species, both ecologically and economically. Its highly adaptive nature allows it to flourish in various environments, as can be seen from its widespread presence from the Arctic and Antarctic seas to the Equator. In natural populations, *Ulva* spp. are very common in littoral and sublittoral areas and also found at mesophotic depths (Spalding et al., 2016; Pyle et al., 2016). As *Ulva* in nature is a holobiome, its ecological role is vast and includes multiple interactions with other players of the marine ecosystems, such as protista, fungi, bacteria, viruses and various marine fauna. *Ulva* is highly relevant for aquaculture due to its fast growth rates and potential food, feed, materials, chemicals and energy applications. Hence, *Ulva* is considered a potential crop for controlled biomass production, onshore and offshore (Fernand et al., 2017). Multiple reports in the last decade addressed *Ulva* aquaculture alone or in multitrophic systems. In addition, *Ulva* biorefinery-enabling processes and technologies have made immense progress in the production of starch, protein, cellulose, ulvan, salts, methane, biocrude, biodiesel, bioethanol, and polyhydroxyalkanoates, just to mention a few (Bikker et al., 2016). Over the years, various systems including plastic sleeves, raceway ponds, tanks, dripping, ropes, nets, rafts and aerated cages have been proposed for *Ulva* biomass cultivation. The variation of cultivation systems ranges from closed, artificial

and seawater, onshore systems with fresh seawater to near shore and far offshore production. Yet, one of the significant risks in *Ulva* cultivation is the sudden biomass loss when the algal tissue disintegrates and bleaches, most probably caused by uncontrolled and unpredicted massive sporulation (Gao et al., 2010; Bruhn et al., 2011).

Opposite to the controlled cultivation, *Ulva* green tides are massive, rapid natural accumulations of unattached green macroalgae biomass usually associated with eutrophicated marine environments.

While green tides of bladed, distromatic *Ulva* species are common (Fort et al., 2020; Zhao et al., 2019; Sfriso et al., 1992; Martins et al., 2001), mass occurrences of monostromatic tubular forms also occur (Blomster et al., 2002; Gao et al., 2010; Cai et al., 2021), which are the focus of this study. The impact of thalli morphology on the potential of *Ulva* species to generate green tides is yet to be determined.

Nevertheless, green tides seriously damage the coastal marine environment, on occasion modifying the shoreline structure, or affecting biodiversity and damaging ecosystem services such as navigation, fishery and recreation (Shan et al., 2019; Zhang et al., 2019). In addition, decaying seaweed biomass causes anoxia yielding hydrogen sulfide at toxic levels in coastal waters and on the shores (Nedergaard et al., 2002; Castel et al., 1996; Viaroli et al., 1995).

Various explanations have been proposed for the rapid accumulation and simultaneous collapse of *Ulva*-dominated green tides. Favorable environmental conditions for *Ulva* habitats, such as temperature, salinity, hydrodynamics and nutrient levels, affect the rapid biomass growth. In addition, recent studies showed that blooming leads to the selection of rapidly growing strains (Fort et al., 2020) with potentially differentially expressed genetic signatures (He et al., 2021). Furthermore, for *Ulva prolifera*, a strain dominating the Yellow Sea bloom, 91.6 %–96.4 % of the released spores developed into young seedlings, suggesting that 1 g fresh weight thallus was able to produce about $2.8 \times 10^8$–$2.7 \times 10^9$ new younger seedlings, of free-floating biomass (Zhang et al., 2013).

*Ulva* sp. has a complex reproduction strategy with alternation of generations in which both isomorphic gametophytes and sporophytes coexist. The gametophytes produce biflagellated haploid gametes through mitosis while the sporophytes produce quadriflagellated haploid zoids through meiosis (Wichard, 2015).

The initiation of a green tide requires simultaneous sporulation and release of gametes of multiple thalli (Gao et al., 2010), but it can be also caused by inhibition of biomass allocation to sporulation (Hiraoka, 2021); the regulation of sporulation is involved in both cases. How this sporulation is achieved and controlled at the initial population is still a puzzling question. The simultaneous release of zoids and gametes in large numbers over a short period, combined with favorable environmental conditions, would provide the prerequisites for the formation of green tides. Indeed, mechani-

cal or other factors, fragmentation of *Ulva* thalli would produce large amounts of spores giving rise to the rapid proliferation of the seaweed under field conditions. This idea was also considered because it likely supports the rapid accumulation of huge biomass of *U. prolifera* in the green tide observed along the Qingdao coasts in 2008 (Gao et al., 2010).

Ecological studies indicate that sporulation in *Ulva* is seasonal, and when it occurs a significant amount of parental biomass contributes to the massive production of swarmers (Amsler and Searles, 1980; Littler and Littler, 1980; Niesenbaum, 1988). The formation and release of swarmers is inhibited by "sporulation-inhibiting substances" excreted into the growth medium by the whole thalli, or their fragments (Nilsen and Nordby, 1975; Jónsson et al., 1985). Later studies identified sporulation inhibitors in *Ulva mutabilis*, *Ulva linza* and *U. prolifera*. The first sporulation inhibitor 1 (SI-1) is a glycoprotein isolated from the thalli media or the cell wall, and the second sporulation inhibitor 2 (SI-2) is a small molecular weight compound that was isolated from the inner space between the two blade cell layers (Stratmann et al., 1996; Jónsson et al., 1985; Kessler et al., 2018; Vesty et al., 2015). Importantly, removing both SIs induces the gametogenesis by washing (and mincing) the algae (Kessler et al., 2018) and activates specific transcription factors (Liu et al., 2022).

Furthermore, formed gametes were only released slowly and asynchronously in the presence of another substance known as a swarming inhibitor, the removal of which resulted in nearly immediate and complete swarming (Stratmann et al., 1996; Wichard and Oertel, 2010). This precise control of swarmer formation and release suggests that *Ulva* developed a tightly regulated mechanism to guarantee simultaneous release of swarmers to the environment, observed initially by Smith (1947) at the Pacific coast (Smith, 1947), probably to maximize the likelihood of sexual reproduction. Indeed, most recent studies on the floating *U. prolifera* showed that all tested thalli were sporophytes with sexual reproductive patterns (Zhao et al., 2019).

Notably, the decrease of SI-1 production coincides with the maturation of *Ulva* and finally causes sporulation (i.e., gametogenesis), first within the *Ulva* apical marginal zones of the thallus and subsequently within the whole alga. Unlike the SI-1, the concentration of SI-2 (per biomass) remains constant, but *Ulva*'s sensitivity towards this inhibitor declines during aging (Stratmann et al., 1996). Unlike SI-2, the presence of SI-1 suppresses gametogenesis at all phases of the life cycle (Fig. 1). In other words, even if *Ulva* no longer produces enough SI-1 during aging, it is still sensitive to external SI-1 application by an aquaculture operator (Fig. 1).

Because only one of the two inhibitors is required to be present and active (Stratmann et al., 1996; Vesty et al., 2015; Kessler et al., 2018), our research concentrates on mathematical model experiments with SI-1, which is produced by the growing *Ulva*, released in excess in its environment and taken up by aging *Ulva*.

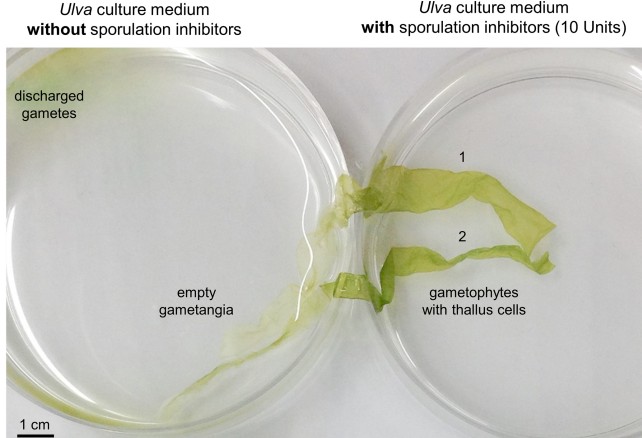

*Ulva* culture medium **without** sporulation inhibitors

*Ulva* culture medium **with** sporulation inhibitors (10 Units)

**Figure 1.** Inhibition of gametogenesis by an externally supplied sporulation inhibitor (SI-1). Two thalli (1 and 2) were transferred into fresh UCM distributed over two Petri dishes. Mature gametophytes undergo gametogenesis upon removal of SIs by washing the thalli. Within three days, gametes were formed and released on the third day after induction (left). The addition of 10 units of SI-1 inhibited the differentiation of thallus cells into gametangia (right) (see Supplement for detailed method description according to Kessler et al., 2018).

Mathematical models are essential tools to study and predict the behavior of complex biological systems. Several models have been developed to predict seaweed biomass growth and decomposition, the behavior of harmful green tides and the seaweed biomass production in seagriculture. Indeed, long-term ecological models that predict macroalgal productivity and seasonal blooms in prone ecosystems (Martins and Marques, 2002; Solidoro et al., 1997; Ren et al., 2014; Martins et al., 2007; Port et al., 2015; Brush and Nixon, 2010; Aldridge and Trimmer, 2009; Lavaud et al., 2020; Seip, 1980; Aveytua-Alcázar et al., 2008; Duarte and Ferreira, 1997) or culture models that focus mostly on onshore photobioreactors (Friedlander et al., 1990; Oca Baradad et al., 2019) and offshore cultivation (Broch and Slagstad, 2012; Petrell et al., 1993; Hadley et al., 2015) were developed. These models, which pursue a basic understanding of the thermodynamics of individual algae thalli and photobioreactors (Zollmann et al., 2018; Lehahn et al., 2016; Martins and Marques, 2002; Lee and Ang, 1991; Seip, 1980), provide important tools to predict the productivity and seasonal environmental effects on the seaweed population dynamics. However, such models treat the macroalgae population as a bulk and do not differentiate between ages of individual thalli within the population. As discussed above, thalli age is an important factor affecting the activity of sporulation inhibitors in the alga and the ultimate release of swarmers. Furthermore, these models do not consider the possible interthalli CEI chemical interactions, some of which can be based on SI-1 secreted to the environment. The production and se-

cretion to the environment of molecules such as SI-1 could provide insights into the molecular mechanisms behind the synchronization of massive spore release at the population level – a phenomenon crucial for both green tide formation and sudden biomass disappearance in *Ulva* seagriculture of species such as *U. mutabilis* (*U. compressa*), *U. linza* and *U. prolifera*.

This paper aims to introduce a novel framework for the description of population dynamics and collective thalli behavior of *Ulva* biomass, presumably controlled by shared sporulation inhibitors. We propose that various environmental and internal biological changes on the single thallus level predetermine the ability of the individual thallus to produce, and to donate to and receive from the population environment, factors that regulate the synchronized formation and release of swarmers. A natural tool to describe this process is offered by population dynamics models, often employed to describe bacterial and animal population dynamics (Succurro and Ebenhöh, 2018; Friedman and Gore, 2017).

In the following sections we develop and simulate such a mathematical model in an attempt to characterize the dynamics of *Ulva* biomass formation and degradation.

## 2 Methods

The model presented below consists of $n + 1$ ordinary differential equations (ODEs), where $n$ is set as the number of cultivation days (also equal to the number of age group equations). ODEs $0 \ldots n-1$ describe the rate of change of biomass of $n$ discrete age classes of *Ulva* thalli, denoted $a_i$, and an additional $n + 1$ equation for the rate of change of the inhibitor $I$. As described previously, there are at least three types of inhibitors involved in the process of *Ulva* swarmers release: SI-1, SI-2 and a swarming inhibitor (SWI). We aggregate these inhibitors into a single quantity that controls the simultaneous swarmers' release from thalli, followed by the biomass decrease.

The $n$ ODEs follow a simple discretized version of a partial differential equation of $m(a, t)$ where $m$ is the biomass of *Ulva* in controlled volume (a bioreactor or a given sea volume, for example), $a$ is the age of algae, and $t$ is time. As previously shown, the growth rate for thalli decreases with age in *U. mutabilis* (Alsufyani et al., 2017).

The following equations specify the dynamics of the biomass of each age class $a_i$, coupled with the dynamics of the inhibitor $I$, in the growth environment:

$$\frac{da_i}{dt} = r_i \left( 1 - \frac{\sum_i^a a_i}{K} \right) + \lambda_i a_{i-1} - \lambda_{i+1} a_i$$
$$- \sigma a_i f(I) \quad i = 1, 2 \ldots n \tag{1}$$

$$\frac{dI}{dt} = \sum_i^n a_i \theta_i - I \left( \sum_i^n \mu_i a_i \right) - \xi I + \gamma_I. \tag{2}$$

The model assumes a logistic growth of biomass with a growth rate parameter $r_i$ for each algae age class $a_i$ and a

carrying capacity $K$, defined as the maximum *Ulva* biomass density in the growth environment. Biomass moves between compartment $a_i$ and $a_{i+1}$ at rates $\lambda_i$, defining the "natural aging" of algae. At each age class $a_i$, biomass is degraded at a rate $\sigma f(I)$, where $\sigma$ is the maximal destruction rate, and $f(I)$ is a monotonically decreasing function of the inhibitor $I$ (SI-1), scaled between 0 and 1. The degradation at a rate $\sigma = 0.3$ is a conservative approximation estimated from a closed aquaculture system with *U. mutabilis* (Alsufyani et al., 2020).

Furthermore, we define $\mu_i$ as the SI-1 uptake rate of each age class $a_i$; $\xi$ the leakage or injection of the inhibitor $I$, which can be managed externally to the system (e.g., washing the algae, destroying the algae, injection nutrients); and $\gamma$ is the nutrient supply flux in the units of inhibitor concentration. Because this is a novel theoretical model, there were no available empirical estimates of the specific functions underlying its dynamics. Hence, whenever a rate was modeled as some monotonic function which saturates, we used the standard in ecological modeling – the logistic equation and exponential decay (Jørgensen and Bendoricchio, 2001). The model functions and parameters are summarized in Table 1.

Finally, we note that our model does not take into account addition of new thalli due to sporulation events during the simulated timeframe. This may be construed as simulating an experiment where spores freely flow out of the container used for growing the algae (e.g., Prabhu et al., 2020). Or this may serve as an approximation if the number of new thalli produced by sporulation in the simulated experimental time frame can be neglected.

## 3   Results and discussion

Here, we study the *Ulva* biomass population dynamics, controlled by sporulation inhibitor production and absorption, by simulating the biomass accumulation under various scenarios for mixed-age populations, for 120 d (Stratmann et al., 1996; Alsufyani et al., 2017). In all the following simulations, we set the initial density of seaweed in the cultivation media to $a_{\text{in}} = 0.2 \, \text{kg m}^{-3}$, and at the maximum carrying capacity the biomass can reach a density of $10 \, \text{kg m}^{-3}$ ($K$). In addition, for the initial population conditions, we denoted young thalli as the population of $a_0$ at $t = 0$ and old thalli as the population of $a_{120}$ at $t = 0$. From a physiological point of view, young thalli are those thalli, whose cell differentiation is controlled by SI and old thalli are those thalli, which do not produce it.

In the following plots, we simulated the behavior of populations with a mixed-age composition. Each population was labeled by the percentage of young and old thalli at initial population. Thus, for example, 100/0 labels an entirely young initial population ($0.2 \, \text{kg m}^{-3}$ of thalli with $a_i = 0$ at $t = 0$); 0/100 labels a completely old initial population ($0.2 \, \text{kg m}^{-3}$ of thalli with $a_i = 120 \, \text{d}$ at $t = 0$); and 50/50

represents an initial population comprised of equal parts of old and young algae ($0.1 \, \text{kg m}^{-3}$ of thalli with $a_i = 0$ at $t = 0$ and $0.1 \, \text{kg m}^{-3}$ of thalli with $a_i = 120 \, \text{d}$ at $t = 0$).

The biomass yield (density increase due to growth; i.e., initial density subtracted from final density) over time for various mixed-age populations is shown in Fig. 2a. The growth of mixed-aged populations with 100/0, 80/20 and 50/50 population mix of old and young thalli showed a typical sigmoidal growth, reaching 90 % of the maximum biomass density ($9 \, \text{kg m}^{-3}$) at 27, 30 and 37 d, respectively. Populations with predominantly old biomass at the beginning (20/80) showed a long lag phase but exhibited positive growth, reaching 90 % of the maximum biomass density at day 87 (Fig. 2b). The population with only old algae (0/100) at the beginning of the cultivation showed degradation of the biomass from day 1 and never showed positive growth (Fig. 2b). The population with a 10/90 mix of initial ages showed a small growth (positive yield) during the first 40 d but then showed biomass degradation and never reached 90 % of the maximum density (Fig. 2a, b).

As simulated in our model, such dynamics of the biomass growth could be explained by the dynamics of the production of the sporulation inhibitor (Fig. 2c). For populations 100/0, 80/20, 50/50 and 20/80 the rate of sporulation inhibitor generation is positive and increases over time, while for the 10/90 and 0/100 populations, although initial production of the sporulation inhibitor is observed, it is reduced over time (Fig. 2c). These findings show that sporulation inhibitor production by a small (20 % in our simulations) young population could potentially provide enough inhibitor to prevent the old algae population from biomass loss, thus leading to overall positive biomass production. Additionally, the rate of biomass accumulation increases with the increased fraction of the young thalli in the initial population. Nevertheless, the maximum inhibitor production decreases with time in all populations as all thalli age (Fig. 2c). To account for a less homogeneous initial population structure, the dynamics were also simulated with a different initial distributions of ages and produced qualitatively similar results (Supplement Fig. S1).

As seaweeds rarely grow in closed bodies of water, where the inhibitor could accumulate continuously in the environment, we sought to stimulate the impact of inhibitor removal from the seaweed environment by water replacement. We simulate each event of water replacement as complete removal of the inhibitor produced by the seaweed during the time interval from the previous water replacement. Figure 3a shows the dynamics of various initial age-mixed populations yields when the inhibitor is removed by water replacement. In the scenario where water is replaced every 14 d, only few, relatively "young" populations (100/0, 90/10, 80/20 and 70/30) achieved the 90 % of the maximum yield. Moreover, the time to reach this yield level increased from 27–30 d (without replacement for these four age groups) to 37–40 d, in the 100/0, 90/10, 80/20 and 70/30 populations, re-

**Table 1.** Model parameters, their interpretation and values.

| Parameter | Meaning | Value |
|---|---|---|
| $r_i$ | Growth rate of algae age $a_i$ | $0.45\left(0.1 + e^{-i\frac{\log(2)}{30}}\right)$ |
| $f(I)$ | Limiting factor due to inhibitor concentration, $I$ | $1 - \left(1 + e^{-10(I-0.5)}\right)^{-1}$ |
| $\sigma$ | Degradation constant | 0.3 |
| $\xi$ | Inhibitor loss function to the environment (for example, water replacement) | $0.45\left(0.1 + e^{-i\frac{\log(2)}{120}}\right)$ |
| $\theta_i$ | Age-dependent inhibitor generation function | $0.45\left(0.1 + e^{-i\frac{\log(2)}{120}}\right)$ |
| $\mu_i$ | Age-dependent inhibitor uptake | $0.45\left(0.05 + e^{-i\cdot\frac{\log(2)}{120}}\right)$ |
| $\gamma_I$ | Constant inhibitor addition or extraction | 0.0–0.1 (varies in the figures) |
| $K$ | Maximal carrying capacity | $10\,\mathrm{kg\,m^{-3}}$ |
| $a_{\mathrm{in}}$ | Initial density | $0.2\,\mathrm{kg\,m^{-3}}$ |

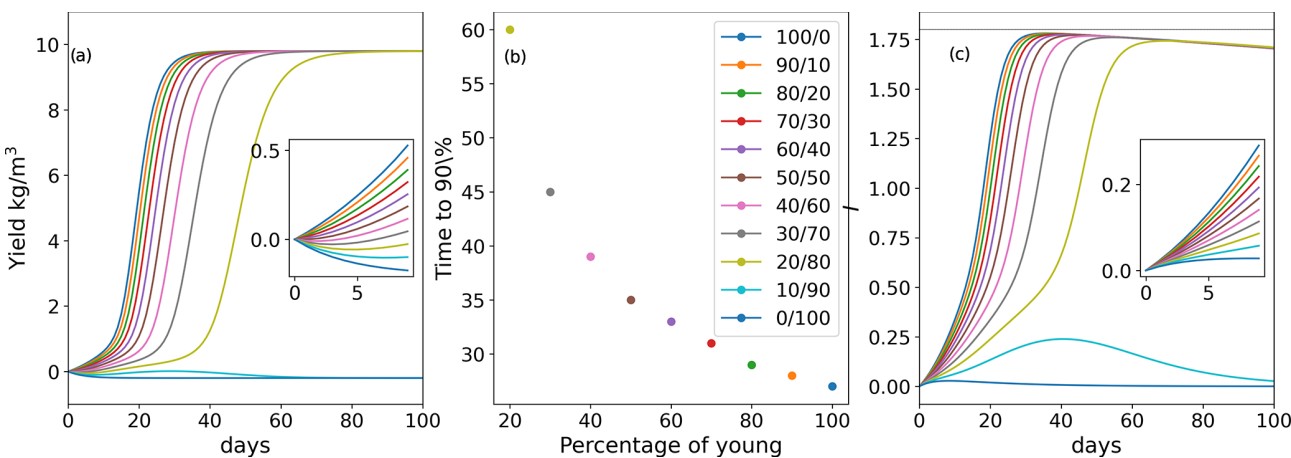

**Figure 2. (a)** Yield for populations with various initial age mixes (biomass gain in $\mathrm{kg\,m^{-3}}$ up to the limit of $10\,\mathrm{kg\,m^{-3}}$ in a bioreactor with a starting density of $0.2\,\mathrm{kg\,m^{-3}}$ in total for each simulated age mix (young/old thalli in %)). **(b)** *Ulva* biomass growth kinetics as a function of the initial age distribution of thalli in the population. Values of the 0/100 and 10/90 populations are not presented as they did not reach 90 % of the maximal carrying capacity. In any other case, mixed populations achieved at least 90 % of the maximal biomass carrying capacity **(c)**. Inhibitor ($I$) amount in the system over time. Insets in **(a)** and **(c)** present zoomed-in dynamics during the first 10 d.

spectively (Fig. 3b). The growth yield shows fluctuating dynamics, showing in the initial overall growth for populations with predominantly young thalli during the first 60 cultivation days, followed by the overall reduction of the yield in the aging populations. Interestingly, these fluctuations yield dynamics that are similar to those previously reported by us during a 12-month offshore cultivation work with *Ulva* harvesting every week (Chemodanov et al., 2018). Although many other factors could be at play, it is possible that the weekly removal of the whole seaweed biomass from the sea and cages for weighting also removed the sporulation inhibitor accu-

mulated in the boundary layer near the thalli. This suggestion, of course, requires further detailed experiments investigating the ability to monitor the dynamics of sporulation inhibitors production and accumulation/diffusion in the thalli environment. In our simulations, increasing the frequency of water exchange, and thus removal of SI-1 (Fig. 3c), reduced the ability of populations with a large portion of old algae to produce a positive yield during the whole cultivation period (Fig. 3a).

As seaweed in the natural environment usually lives in high-energy conditions, we also studied the coupled effects

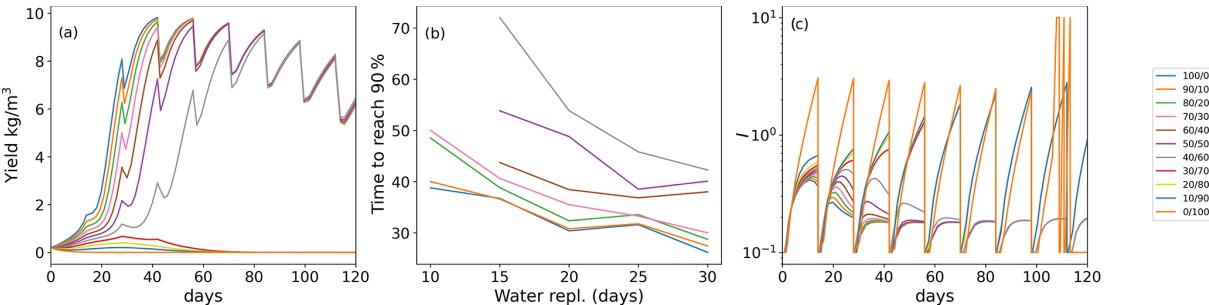

**Figure 3. (a)** Yield (biomass gain in $kg\,m^{-3}$ up to the limit of $10\,kg\,m^{-3}$ in a bioreactor with a starting density of $0.2\,g\,m^{-3}$) for populations with various initial age mixes (% of young/old thalli) with 14 d water replacement frequency. **(b)** Inhibitor ($I$) production in the population of the time with 14 d of water replacement frequency. **(c)** *Ulva* biomass growth kinetics as a function of the initial age distribution of thalli in the population with various frequencies of water replacement.

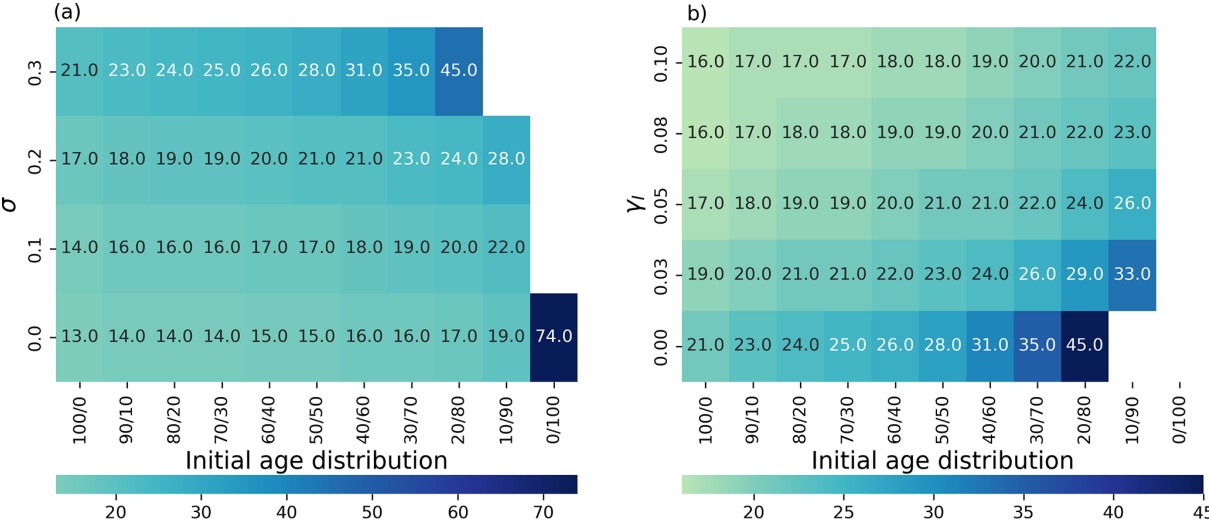

**Figure 4. (a)** Time to achieve 90 % of the maximum carrying capacity as a heatmap (color represents the time in days) depending on the initial age distribution ($x$ axis; (% of old/young thalli)) and the degradation parameter $\sigma_i$ ($y$ axis). **(b)** Time to 90 % as a heatmap (color represents the time in days; note the different scales in **a** and **b**) depending on the initial age distribution ($x$ axis) and the addition of the external inhibitor $\gamma$ ($y$ axis). White color means that the population never achieves 90 % of the maximum carrying capacity.

of degradation ($\sigma$) and population age distribution on the ability of the population biomass yield (Fig. 4a). Higher rates of degradation prevent positive yields in all mixed-age populations. Lower degradation affects a smaller portion of the population with higher initial portions of young thalli (Fig. 4a) showing again the regenerative ability of the populations with high growth (lower sporulation) capabilities.

We investigated the impact of direct addition of the sporulation inhibitor to the seaweed growth media computationally (Fig. 4.b). Adding an external sporulation inhibitor (up to 0.1) reduced the time to achieve 90 % of the maximum yield from 27 d (without inhibitor addition) to 21 d for 100/0 group, 29 d (without inhibitor addition) to 24 for 80/20 group, from 35 d (without inhibitor addition) to 28 for the 50/50 group and from 60 (without inhibitor addition) to 45 d for the 20/80 group. No effects at this maximum concentration have been observed for the 0/100 group.

## 4 Conclusions

In this study, we aimed to further understand the growth and sporulation dynamics of *Ulva* using a mathematical model. We found that successful accumulation of *Ulva* biomass depends on the age distribution of the *Ulva* thalli of a given seed stock, where older starter populations produce lower yields at a given time. However, this age-dependent effect can be mitigated, leading to prolonged maintenance of *Ulva*'s aquacultures, by external addition of sporulation inhibitors. We note that our modeling study is a first attempt to uncover the mechanism underlying the heterogeneity of vegetative growth stability in different *Ulva* populations. At this point, we developed a general model, which is not species specific. Indeed, several studies have revealed that the SIs of *U. linza*, *U. compressa* (*U. mutabilis*) and *U. intestinalis* are exchangeable (Vesty et al., 2015; Steinhagen et al., 2019; Stratmann

et al., 1996). We assume the more closely related the *Ulva* species are, the more likely the SIs will be interchangeable between the species. The SI-1, in fact, cannot be swapped between *U. compressa* and *U. rigida* (Stratmann et al., 1996).

Bioassays with SI-1 have supported the finding of the model that age-mixed populations are more stable than uniform ones. When purified SI-1 is added to mature and induced *Ulva* thalli, it can be still perceived by *Ulva* and used to regulate gametogenesis such in the case of aging cultures of the 0/100 group. We thus conclude that the sporulation event is delayed or even inhibited in age-mixed cultures composed of young, smaller thalli and old, larger thalli.

Overall the sporulation phenomenon creates unique constraints on the age structure of *Ulva* populations. In higher plants, sexual reproduction and vegetative propagation compete for nutrients, but the competition may be mitigated by separating these processes through time (Evans and Black, 1993). However, in *Ulva*, the whole thallus can be transformed into gametangia and sporangia while flowering plants assign only a specific portion of biomass to reproductive structures. *Ulva* thus requires a strict regulation of sporulation, e.g., through the age-dependent production of SI. Only if the SI-1 synthesis ceases during the *Ulva*'s development cycle and its concentration falls below a critical threshold concentration, gametogenesis is induced at positions of the blade where the SI-2 concentration between the cell layers is also sufficiently low or not perceived anymore (Stratmann et al., 1996). Our findings thus imply that the more SI provided by young algae in mixed cultures, the higher the growth rate and biomass yields. As purified SIs are not yet widely available in large quantities, the use of mixed-aged cultures can be an important tool to maintain them at adequate nutrient levels, e.g., in integrated multi trophic aquaculture. The modeling of *Ulva*'s growth indicates the importance of SI-producing algae for sustainable and successful seagriculture and paves the way for a better understanding of the green tide formation in coastal areas.

*Code availability.* The model creation process is explained in detail in the Methods. Code for creating the figures is available upon request.

*Data availability.* All the analyses in the paper can be recreated through the code deposited here: https://github.com/alexliberzonlab/mixed_age_algae_population_modeling/tree/1.0 (last access: 6 April 2022, Obolski and ALiberzon, 2022). TS1

*Supplement.* The supplement related to this article is available online at: https://doi.org/10.5194/bg-19-1-2022-supplement.

*Author contributions.* AG and AL conceived the initial idea for the study; UO, AG and AL designed the study and the mathematical model; AL produced the results; TW produced the experimental results; all authors interpreted the results and wrote the manuscript.

*Competing interests.* The contact author has declared that neither they nor their co-authors have any competing interests.

*Acknowledgements.* Ralf Kessler (Friedrich Schiller University Jena, Germany) is acknowledged for carrying out the shown biotest with *Ulva*.

*Financial support.* The authors thank the Israel Ministry of Health (grant no. 3-16052) for the support. This article is based upon work from COST Action CA20106, supported by COST (European Cooperation in Science and Technology, https://www.cost.eu, last access: 31 March 2022). This research has been supported by the Deutsche Forschungsgemeinschaft through grant no. SFB 1127/2 ChemBioSys–239748522 (Thomas Wichard).

*Review statement.* This paper was edited by Andrew Thurber and reviewed by Erik-jan Malta and one anonymous referee.

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

**Remarks from the language copy-editor**

CE1     The hyphen is correct here, as inter- is a prefix, not a word in its own right.

**Remarks from the typesetter**

TS1     Please confirm.
TS2     Please confirm.