# Peer review of "Modeling the growth and sporulation dynamics of the macroalga *Ulva* in mixed-age"

_Biogeosciences, 2021_

## Author Comment (AC1)

Dear Editor,

Thank you very much for considering a revision of our manuscript titled "Modeling the growth and sporulation dynamics of the macroalga *Ulva* in mixed-age populations in cultivation and the formation of green tides" (**BG-2021-183**) for *Biogeosciences*.

Please find the reviewers' comments (regular font) with a point-by-point response (**in bold**). We have answered all the points raised and hope that you now find the manuscript suitable for publication.

**Anonymous Referee #1**

This is an interesting, generally succinct, and strongly written contribution focusing on mathematical modeling of *Ulva* sporulation and growth. *Ulva* seagriculture is an emerging industry, thus this modelling is relevant both in terms of IMTA and the occurrence of green tides.

**We thank the reviewer for the kind words.**

My primary issues with the paper focus on two aspects: 1) the graphics – Figures 1 and 2 need to be made bigger to better demonstrate the patterns illustrated. I'm also uncertain as to the utility of Figure 3 – it mostly just looks blue, with a small hint of yellow. Either make this figure more meaningful with a better spread of color differentiation, or remove.

We agree. Figures 1 and 2 (now Figures 2 and 3) have now been uploaded at a higher resolution. As for Figure 3 (now Figure 4), we have now changed the color scale and added a number to each square for clarity.

2) Verification - this paper focuses on just the model, and doesn't provide any application of the model to real world data. While there is one instance where the authors note that the model appears to be consistent with what they previously observed, I felt myself wanting verification that these models are accurate. For instance, if you apply these models to the real world, would the models verify what actually happens in the field? I would like to see this issue addressed in more detail. For instance, do the authors have any field studies that can be added to this contribution, or is this the next step?

We thank the reviewer for their important comment. We should have clarified this better in the manuscript: This is a theoretical study, laying the foundations of future empirical research. We started with a bottom-up, mechanistic, approach of modelling based on the information from the literature. As the reviewer mentioned, we also provide a small example in our SI. This example and supporting text have now been moved to the main text and are explained more elaborately (revised Results and Discussion section).

Furthermore, we now more clearly discuss a specific in-situ example (with all its shortcomings):

"Interestingly, these fluctuations yield dynamics that are similar to those previously reported by us during a 12-month offshore cultivation work with *Ulva* harvesting every week (Chemodanov et al., 2018). Although many other factors could be at play, it is possible that the weekly removal of the whole seaweed biomass from the sea and cages for weighting also removed the sporulation inhibitor accumulated in the boundary layer near the thalli. This suggestion, of course, requires further detailed experiments investigating the ability to monitor the dynamics of sporulation inhibitors production and accumulation/diffusion in the thalli environment."

**Comments by line:**

33: note that Ulva can also be found at mesophotic depths, and not just the littoral and sublittoral zones

**We thank the reviewer and have corrected the sentence with relevant references.**

38 – 47: These sentences are missing citations for the information presented

**We have now added several references to this section to better support our statements.**

40: change "in producing" to "the production"

**Changed.**

48 – 53: These sentences are also missing citations for the information presented

**We have now added several references to this section to better support our statements.**

202 - 203: change "could happen" to "is possible"

**Changed.**

236: change "algae production" to "algal production"

**We believe you meant "population" rather than "production". Changed.**

**Reviewer: Erik-jan Malta:**

**General comments**

Interesting paper simulating potential effects of the effect of sporulation dynamics related to thallus age on biomass build-up of the green macroalga Ulvasp. The mechanism and dynamics of sporulation in this macroalgal genus are still largely unraveled. The model presented in the manuscript illustrates their importance both in nature in Ulvablooms as in applications such as large scale Ulva cultivation – a highly relevant issue nowadays.

In my opinion, the model should be considered as a a theoretical test of the hypothesis that differences in sporulation between differently aged Ulva can influence biomass build-up in a population. Growth and thus biomass build-up is influenced by many external factors, starting with temperature, light and nutrient availability that may even lead to periodic patterns (for instance nutrient supply with spring tides, etc.). In addition, due to biomass build-up, algae will experience reduction of light availability due to self-shading, increased competition for nutrients, both directly and due to reduced water flow, increasing pH leading to reduced carbon uptake, etc. that might also generate these patterns, especially when combined with periodic biomass export (as in cultures for examples). This model is too simple to interpret actual data as the authors suggest (lines 201-206), however tempting this may be. Nevertheless, bearing this in mind, in my view the authors convincingly demonstrate the importance of population structure on growth dynamics, especially under articial conditions (cultivation) where other factors such as nutrients are controlled.

**We thank the reviewer for providing us with the review and for his interest in our work. We completely agree that this is a theoretical model that serves as a proof of concept and is intended for hypothesis generating and setting the stage for future work.**

Nevertheless, there are still a number of points that require clarification and possibly correction, as it appears to me that there are some points in which the authors contradict themselves. Starting with the latter, in lines 93-95 it says "Additional studies showed that if the thalli are ageing (i.e. no

growth) they become insensitive to the artificially added sporulation inhibitor and release swarmers even in its controlled presence in the growing media (Alsufyani et al., 2017)." In their model, the authors define as old algae "those thalli, which are insensitive to SI and do not produce it." (I. 157-159). In that light I do not understand the explanation of the model outcome offered in I. 180-182: "These findings show that sporulation inhibitor production by a small (20% in our simulations) young population could potentially provide enough inhibitor to prevent the old algae population from biomass loss..." This seems like a contradiction to me – the old algae should be completely insensitive to the inhibitor as the authors stated earlier.

Thank you for the comment. Our phrasing was misleading. We have now clarified the different modes of action of the SIs in the text:

"Notably, the decrease of SI-1 production coincides with the maturation of *Ulva* and finally causes sporulations (i.e. gametogenesis), first within the *Ulva* apical marginal zones of the thallus and subsequently within the whole alga. Unlike the SI-1, the concentration of SI-2 (per biomass) remains constant, but *Ulva*'s sensitivity towards this inhibitor declines during aging (Stratmann et al. 1996). Unlike SI-2, the presence of SI-1 suppresses gametogenesis at all phases of the life cycle (Fig. 1). In other words, even if *Ulva* no longer produces enough SI-1 during aging, it is still sensitive to external SI-1 application by an aquaculture operator (Fig. 1). Because only one of the two inhibitors is required to be present and active (Stratmann et al. 1996, Vesty et al. 2015; Kessler et al. 2018), our research concentrates on mathematical model experiments with SI-1, which is produced by the growing *Ulva*, released in excess in its environment and taken up by aging *Ulva*."

The model has been based on mixtures of two "generations" of algae. Even though they "age" in the model, there is always only two generations. If I understand the model correctly, the new young algal biomass that will arise as a consequence of the sporulation is not accounted for. Although I appreciate the complexity of incorporating this in the model, I think some speculation would be in place on the potential effect of this, as it probably slows down population aging and its negative effects.

We thank the reviewer for bringing this point to our attention - we have not explained well enough in the text. In our model, we simulate an environment for the growth of a single 'batch' of algae, in a container with sufficient outflux for spores to be washed away. E.g. a net holding algae at sea. Hence the sporulation events should not affect the new algal biomass. On the other hand, our simulations can approximate situations where the effect of new thalli is negligible in the relevant timeframe. This is now explained better in the text:

"Finally, we note that our model does not take into account addition of new thalli due to sporulation events during the simulated timeframe. This may be construed as simulating an experiment where spores freely flow out of the container used for growing the algae (e.g. (Prabhu et al., 2020)). Or, this may serve as an approximation if the amount of new thalli produced by sporulation in the simulated experimental time frame can be neglected."

Finally, in the simulated experiments, we set a proportion p of the population at age class 0 and 1-p at age class 120, to emulate a fraction of young algae. All the young algae age throughout the simulations and some of them reach the 120 class, so there are no two distinct generations. However, this was an arbitrary choice of a distribution for the convenience of presentation. We have now added to the SI plots where algae are exponentially distributed around different initial ages and get qualitatively similar results (see SI figure S2). This is also mentioned in the text.

Another point of concern are the parameter values selected for the model. The paper does not list a literature or other basis for these values, nor has a sensitivity analyses been performed to assess the effect of changes in them. For instance, if I understand correctly, the degradation constant is the biomass lost, supposedly in part due to sporulation, set at 0.3, or 30% of the biomass at maximum in the oldest age class. This seems quite arbitrary and maybe quite conservative some reference mention complete loss of biomass due to sporulation. What would be the effect if this factor was much higher, for instance 0.9? The same goes for the effect of algae age – are the response equations based on experimental data or are these choices of the modelers?

We now provide further details for the parameter value of the degradation constant  $\sigma$  used in figure 2 and 3. The value 0.3 was estimated from an experiment with 200 L Ulva tanks performed at the Ramalhete Station of the Centro de Ciências do Mar (CCMAR) in Faro (Portugal). The authors are aware that the degradation constant (due to sporulation and fragmentation) is a conservative proxy (Alsufyani et al. 2020). We now added this to the main text.

"The degraded at a rate  $\sigma$  = 0.3 is a conservative proxy estimated from a closed aquaculture system with *U. mutabilis* (Alsufyani et al., 2020)."

Importantly, the values of the degradation constant  $\sigma$  (as well as the rates of the external inhibitor addition) are varied in Figure 4.

Finally, since this is a theoretical model with a lot of unknowns, there have to be some simplifying choices. We now explain in further detail that all of our growth/inhibition functions were chosen by the most standard ecological modeling function of saturating rates - the logistic function and exponential decay:

"Since this is a novel theoretical model, there were no available empirical estimates of the specific functions underlying its dynamics. Hence, whenever a rate was modeled as some monotonic function which saturates, we used the standard in ecological modeling - the logistic equation and exponential decay (Jørgensen and Bendoricchio, 2001). The model functions and parameters are summarized in Table 1."

**Specific comments**

I. 47 – This merits a reference.

**Added - Gao S, Chen X, Yi Q, Wang G, Pan G, et al. (2010) A Strategy for the Proliferation of *Ulva prolifera*, Main Causative Species of Green Tides, with Formation of Sporangia by Fragmentation. PLOS ONE 5(1): e8571. https://doi.org/10.1371/journal.pone.0008571**

I. 50 – The species mentioned are all filamentous, the Enteromorpha morphology. In Europe blooms mainly consist of foliose Ulva species. Biomass losses due to sporulation might be less important in those.

Some of the literature on European eutraphican also mentions the same filamentous species (we added the references to the text). We do agree with the reviewer that there is a need to study the impact of morphology on the sporulation of tide formation potential of

species (distromatic sheets *versus* monostromatic tubes). We added this comment to the paper. However, the tubular *Ulva compressa* contributes to green tides in Yellow Sea (China) (e.g. Cai et al. 2021) and the Baltic Sea in Europe (Blomster et al. 2002).

In any case, biomass loss due to sporulation can be also observed in aquacultures of folios *Ulva*. In fact, the low biomass of *U. lactuca* recorded in aquacultures was the result of sporadic sporulation events in some of the operated tanks (Bruhn et al. 2011).

I. 67 – I do not agree with this (see also above). For foliose Ulvagreen tides, bloom initiation is thought to origin from vegetative fragments, see Kamermans et al. (1998) Mar Biol 131: 45-51 and other authors.

The initiation of a green tide requires simultaneous sporulation and release of gametes of multiple thalli (Gao et al. 2010), but can be also caused by inhibition of biomass allocation to sporulation (Hiraoka, 2021). In any case the regulation of sporulation is involved.

I. 73 – The Qingdao bloom did not only occur in 2008 but seems to have converted into an annual phenomenon. In addition, although sporulation might be certainly accelerate growth, the basic requirement for the rapid accumulation is the high nutrient availability.

The sentence has been rephrased: "This idea was also considered since it likely supports the rapid accumulation of huge biomass of *U. prolifera* in the green tide observed along the Qingdao coasts in 2008 (Gao et al., 2010)."

I. 93 – Has this been demonstrated for other Ulvaspecies as well?

These sentences have been completely rephrased as also indicated by the first reviewer. Sls were studied on *U. mutabilis*, *U compressa*, *U linza* and *U. prolifera*.

I. 104 – Basic understanding of thermodynamics only refers to Zollmann et al. 2018.

**Additional references were added.**

I. 113 – This is not necessarily the case for all Ulva species. Some, such as U. ohnoi in south Spain and Portugal never show sporulation. For other areas, such as the foliose Ulva blooms in Brittany, France, and Venice, Italy several authors suggested other causes for biomass degradation. Sporulation can certainly important, but this might be species specific within the genus.

Thanks, for this important comment. Yes it is true. We have specified the particular species. However, future studies will study the regulation of SI-production and provide explanations why the level is always high in case of *U. ohnoi* and drops down the threshold concentration in certain habitats.

I. 134 – How general is this in Ulva, is this process linear, experimental data?

**The sentence was rephrased.**

I. 144 - f (I) is monotoning decreasing function? I am not too experienced in modeling so I might be mistaken, but should this not be increasing from 0 to 1 so as to maximize degradation in the oldeste age class?

f(I) is indeed a monotone decreasing function of I, and in the differential equations the term for degradation appears as f(I), multiplied by a scaling factor sigma, and a minus sign thus degradation is highest when I is lowest, as you expect. The same degradation rate appears in all a\_i. The difference is in the decreasing growth rate with age: r\_i, and when this rate is lower than the degradation rate, the algae stops growing.

I. 153 – "We assumed", unless it is based on literature or experimental data this is a setting, not an assumption.

**Fixed, thanks.**

I. 158 – However, due to the sporulation, new t0 generations are constantly formed and even more so in older populations where the rate is maximum. This might dampen the modeled curves.

**We fixed the text and addressed it in response to your main comments.**

I. 192 – Again this is not an assumption but a setting.

**Fixed, thanks again.**

I. 202 and further – as argued in the general comment, this is highly speculative and might very well be a coincidence. Weekly biomass removal has a number of effects with respect to light and nutrient availability as well in addition to the natural variation in parameters as light and temperature.

**We agree and have mitigated the text in this section.**

I. 207 – Contradictory to earlier statement that older algae are insensitive to sporulation inhibitors.

We have clarified this statement. *Ulva* is insensitive to SI-2 during aging but not to SI-1, which is under investigation of this study. In other words, although *Ulva* does not produce enough SI-1 anymore it is still sensitive to the inhibitor.

I. 238 – Again, this seems like a contradiction to me. If old algae are indeed insensitive to sporulation inhibitors, adding them from a certain population age will not have an effect.

We fixed the text and addressed both these comments in response to your main comments.

I. 252 - At least for some Ulva species, not clear how species-specific this is.

At this point this is a general model and is not species specific. In fact, up to now it was shown that the sporulation inhibitors can be easily exchanged between *U. linza, Ulva compressa (U. mutabilis)* and *Ulva intestinalis* (Stratmann et al. 1996; Steinhagen et al. 2019); Vesty et al. 2015). In fact SI-1 is not exchangeable between *U. compressa* and *U. rigida* (Stratmann et al. 1996). We assume that the closer the relationship between the *Ulva* species, the more likely it is that the SIs are exchangeable. It was demonstrated that *U. lactuca* and *U. prolifera* are producing those inhibitors as well (Wichard and Oertel 2010; Jónsson et al. 1985), but an exchange experiment with those species mentioned above was not performed up to now.

In summary, we are confident that there will be variation between species, but this could be delineated only experimentally. We now mention this point in the Conclusions:

"We note that our modelling study is a first attempt to uncover the mechanism underlying the heterogeneity of vegetative growth stability in different *Ulva* populations. At this point, we developed a general model, which is not species specific. Indeed, several studies have revealed that the SIs of *U. linza*, *U. compressa* (*U. mutabilis*), and *U. intestinalis* are exchangeable (Vesty et al., 2015; Steinhagen et al., 2019; Stratmann et al., 1996). We assume the more closely related the *Ulva* species are, the more likely the SIs will be interchangeable between the species. The SI-1, in fact, cannot be swapped between *U. compressa* and *U. rigida* (Stratmann et al., 1996)."

**Technical corrections**

I. 191 – "stimulate" should be "simulate"

**Fixed, thanks.**

Citation Wichard 2015 is lacking in the bibliography.

**Fixed, thanks.**